

# Extreme ozone episodes in a major Mediterranean urban area

Jordi Massagué[1,2], Eduardo Torre-Pascual[3], Cristina Carnerero[1,4,*], Miguel Escudero[5], Andrés Alastuey[1], Marco Pandolfi[1], Xavier Querol[1], Gotzon Gangoiti[3]

[1]Institute of Environmental Assessment and Water Research (IDAEA-CSIC), Barcelona, 08034, Spain
[2]Department of Mining, Industrial and ICT Engineering, Universitat Politècnica de Catalunya-BarcelonaTech, (UPC), Manresa, 08242, Spain
[3]Faculty of Engineering Bilbao. University of the Basque Country (UPV/EHU), Bilbao, 48013, Spain
[4]Department of Civil and Environmental Engineering, Universitat Politècnica de Catalunya-BarcelonaTech, (UPC), Barcelona, 08034, Spain
[5]Department of Applied Physics. School of Engineering and Architecture. Universidad de Zaragoza (UNIZAR), Zaragoza, 50018, Spain
*Current affiliation: Barcelona Supercomputing Centre, Barcelona, 08034, Spain

*Correspondence to:* Jordi Massagué (jordi.massague@idaea.csic.es)

**Abstract.** This study analysed three extreme ozone ($O_3$) episodes that occurred in Barcelona (NE Spain) during the summers of 2015, 2018, and 2019. These episodes were the only instances since at least the year 2000 when the EU's hourly information threshold (180 µg·m$^{-3}$) was exceeded in the city. Understanding the phenomenology of these episodes is crucial owing to Barcelona's large population. Using experimental data and diverse modelling tools, the main objective was to elucidate the underlying phenomena of recent extreme $O_3$ episodes, identify shared patterns and improve future predictions. The results revealed a complex interplay of factors contributing to the episodes, including (i) initial regional $O_3$ accumulation, (ii) the potential impact of Tramontana winds on $O_3$ transport into the western Mediterranean basin, (iii) $O_3$ accumulation via vertical recirculation and horizontal circulation of the local pollution plume in a weakened sea breeze regime, (iv) convergence of polluted air masses from multiple sources, (v) calm upper layer winds, (vi) the weekend effect, and (vii) abnormally high temperatures. Some of these factors, which may manifest in the days preceding the episodes, are readily observable or can be anticipated. Consequently, the findings of this study enhance the understanding of the mechanisms driving these extreme $O_3$ episodes and provide valuable insights for their prediction.

## 1. Introduction

Tropospheric ozone ($O_3$) is a strong oxidising secondary atmospheric pollutant, adversely affecting human health, ecosystems and materials (WHO, 2006, 2013a, b; GBD, 2016; Fowler et al., 2009; IPCC, 2021). Its formation depends primarily on precursors (NOx, CH$_4$, CO and non-methane volatile organic compounds (VOCs)) and solar radiation. Its production is enhanced by high temperatures and low relative humidity (Monks et al., 2015), contributing to extreme $O_3$ events. These events





have been linked to heat waves and increased mortality globally (e.g., Vautard et al., 2007; Guo et al., 2017; Pu et al., 2017; Jaen et al., 2021).

The NOx/VOC ratio considerably influences $O_3$ formation. $O_3$ production is typically VOC-limited in urban environments
owing to relatively high NOx levels, while it is typically NOx-limited in suburban and rural environments where NOx/VOC levels are low (Sillman et al., 1999; Sillman and He, 2002). In urban environments, the VOC-limited regime typically results in the "weekend effect", where $O_3$ concentrations are typically higher on non-working days. This is due to lower emissions of $O_3$-depleting pollutants (Heuss et al., 2003; Jiménez et al., 2005).

Epidemiological studies have indicated detrimental impacts on human health from both long-term and short-term exposure to
$O_3$ (WHO, 2021). In some European regions, $O_3$ pollution presents a major air quality (AQ) problem (EEA, 2015), leading to the establishment of AQ standards to mitigate its harmful effects. The European AQ Directive 2008/50/EC (EC, 2008) has set standards for chronic and episodic $O_3$ exposure.

Southern Europe, particularly the Mediterranean, faces substantial $O_3$ pollution (EEA, 2020). In the western Mediterranean basin (WMB), many factors influence high $O_3$ concentrations (e.g., Millán et al., 1997, 2000; Gangoiti et al., 2001; Millán,
2014), including characteristic meteorological, climatic and topographic patterns, high biogenic emissions during hot seasons (Seco et al., 2011), recurrent mesoscale circulations during summer (Diéguez et al., 2009, 2014), regional and hemispheric $O_3$ transport (Pay et al., 2019), high emissions of precursors in specific atmospheric basins (Querol et al., 2017, 2018; Escudero et al., 2019) and stratospheric intrusions (Kalabokas et al., 2017). $O_3$ contributions to surface concentrations can vary widely in time and space. Hence, the causes of AQ standard exceedances can vary considerably, even within a given basin. Thus, $O_3$
concentrations can result from (i) local formation from precursors emitted in the atmospheric basin, which can be favoured by complex vertical recirculation of air masses; (ii) regional transport from other air basins in Spain and other parts of Europe; (iii) hemispheric transport; or (iv) stratospheric intrusions.

Concentrations of $O_3$ in Barcelona (NE Spain) are not particularly higher than in other regions of the country (Querol et al., 2016; Massagué et al., 2023) owing to strong $O_3$ titration and ozonolysis (Monks et al., 2015) caused by local emissions of
NO and VOCs, resulting in infrequent exceedances of the legal thresholds set by the European AQ Directive. However, since at least 2000, exceedances of the hourly Information Threshold (IT, $O_3$ concentration >180 $\mu g \cdot m^{-3}$) have only been detected in 2015, 2018 and 2019 (Gencat, 2022). These extreme events exposed a large population segment to exceptionally high $O_3$ concentrations, which is particularly concerning as Barcelona is Spain's second most populous urban area, with the highest population density neighbourhoods in Europe (Batista et al., 2021).

This study aimed to elucidate the underlying phenomena behind the recent episodes of extreme $O_3$ concentrations in Barcelona, determine potential common patterns and obtain relevant information to enhance the prediction of future occurrences.



## 2. Methodology

To comprehensively analyse the episodes, we propose a combined methodology that integrates multiple approaches and datasets, including ground-based and satellite observational data, meteorological reanalysis, high-resolution backward and
forward trajectory simulations and photochemical simulations.

### 2.1 Study area

This study focused on Barcelona, Spain's second most populous municipality, located on the NE coast of the Iberian Peninsula (IP), with 1.6 million inhabitants within the city limits and up to 5.2 million in neighbouring urban areas (MITMA, 2021). Barcelona is located on a 170 km$^2$ coastal plain bordered by the Mediterranean to the east, with the Collserola mountain range
(up to 516 m above sea level (ASL)) to the NW and two river mouths to the N and SW (Besòs and Llobregat, Fig. 1b). All influence local airflow patterns (Toll and Baldasano, 2000). This orography induces mesoscale circulations, such as sea/mountain breezes channelling northward towards the Pyrenees (e.g., Barros et al., 2003; Diéguez et al., 2009).

Barcelona and its surrounding metropolitan area (BMA) are significant sources of NOx and VOC emissions from road traffic, industry, shipping activities and domestic sources. In summer, air masses loaded with $O_3$ and its precursors are transported
inland and, combined with high biogenic VOC emissions from nearby forested areas, may cause severe $O_3$ episodes in downwind areas (e.g. Toll and Balsano, 2000; Pérez et al., 2004; Gonçalves et al., 2009; Diéguez et al., 2009; Valverde et al., 2016; Querol et al., 2017).

Diéguez et al. (2009) proposed three routes for emissions from the BMA and surroundings to reach the Oriental Pyrenees (Fig. 1c), (i) the Llobregat basin (Llobregat-axis), (ii) the Besòs–Ter basin (Besòs-axis), and (iii) the NE-axis between the coastal
and pre-coastal ranges, which includes a major road with high traffic emissions (AP-7 highway). Air masses can flow along these routes, with meteorological conditions favouring one over others (Diéguez et al., 2009). These transport patterns lead to frequent exceedances of $O_3$ thresholds at air quality monitoring stations (AQMSs) along these axes. A representative example is the Vic Plain, located in the Besòs-axis, 60 km N of Barcelona, which has been a hotspot for IT exceedances in Spain (Querol et al., 2016; Massagué et al., 2023).

Similar to the BMA, emissions of industrial and urban pollutants from Tarragona contribute to the typical high $O_3$ levels in the northern and northwestern areas Querol et al., 2016). Tarragona is located on the coast 90 km SW of Barcelona (Fig 1b), and it is surrounded by notable chemical and petrochemical industrial complexes.



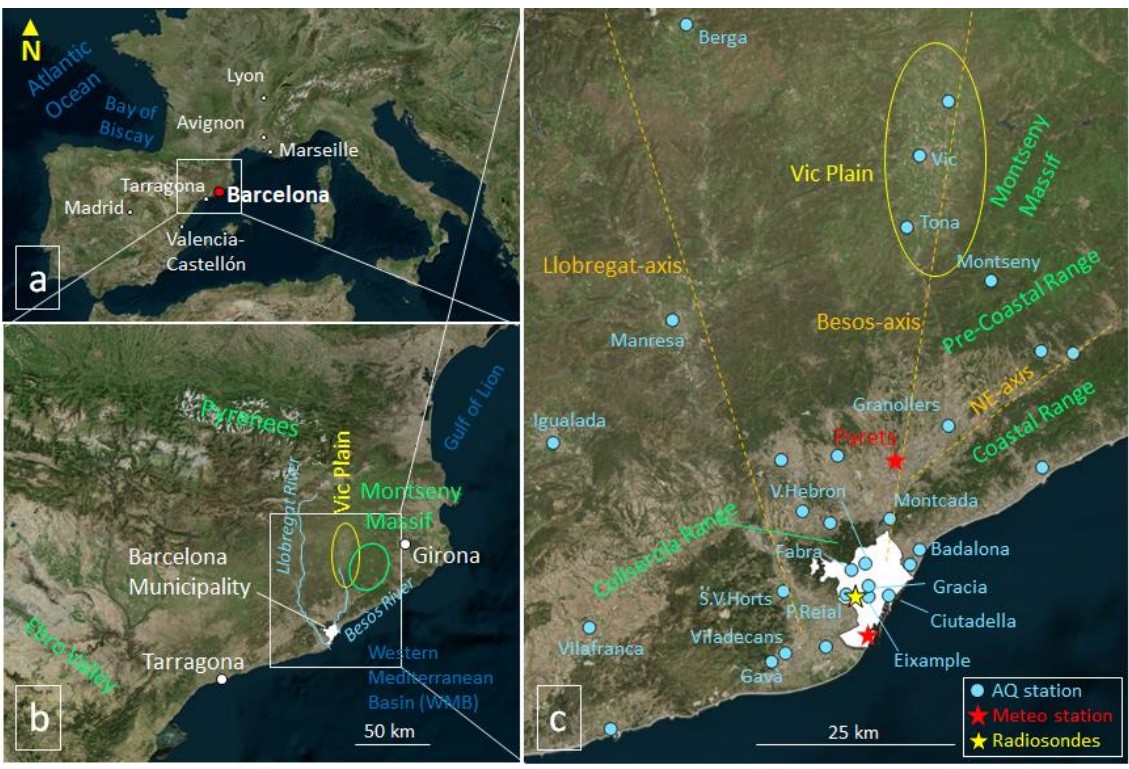

| Localisation | Name of AQMS | Type | European code | Latitude (deg.) | Longitude (deg.) | Altitude (m above sea level) |
|---|---|---|---|---|---|---|
| Barcelona | *Port* | *Meteo* | *D5* | *41.31725* | *2.16537* | *3* |
| | *Faculty of Physics Univ. Bcn* | *Radiosondes* | *-* | *41.38553* | *2.11720* | *70* |
| | Ciutadella | UB | ES1679A | 41.38641 | 2.18742 | 7 |
| | Fabra | SB | ES2090A | 41.41840 | 2.12390 | 418 |
| | P.Reial | UB | ES1992A | 41.38748 | 2.11515 | 81 |
| | Gràcia | UT | ES1480A | 41.39874 | 2.15339 | 57 |
| | V.Hebron | UB | ES1856A | 41.42608 | 2.14799 | 136 |
| | Eixample | UT | ES1438A | 41.38534 | 2.15382 | 26 |
| Barcelona Metropolitan Area (BMA) | Badalona | UB | ES1892A | 41.44398 | 2.23788 | 7 |
| | Gava | SB | ES1910A | 41.30311 | 1.99152 | 25 |
| | Viladecans | SB | ES1903A | 41.31348 | 2.01382 | 14 |
| | S.V Horts | SB | ES0694A | 41.39216 | 2.00980 | 38 |
| | Montcada | ST | ES0584A | 41.48202 | 2.18830 | 34 |
| | Granollers | UT | ES1891A | 41.59867 | 2.28712 | 133 |
| | *Parets* | *Meteo* | *XG* | *41.56734* | *2.22619* | *123* |
| Montseny Massif | Montseny | RB | ES1778A | 41.77934 | 2.35802 | 693 |
| Vic Plain | Tona | RB | ES1923A | 41.84603 | 2.21758 | 620 |
| | Vic | SB | ES1642A | 41.93567 | 2.23857 | 498 |

**Figure 1. Top: Area of study. Blue dots represent air quality monitoring stations (AQMSs). Only names of AQMSs used in the study are displayed in (c); others are kept for reference. Red stars indicate meteorological stations. The yellow star indicates the location of radiosounding launches. Bottom: Main characteristics of the AQMSs. Types are UB−urban background, UT−urban traffic, SB−suburban background, ST−suburban traffic and RB−rural background. Meteorological stations are identified with their**



**meteorological office's code. For Barcelona, wind data are from the Barcelona port, and temperature and relative humidity data**
**are from the same location as the AQMS Fabra. The white-shaded area depicts the municipality of Barcelona.**

Gangoiti et al. (2001, 2006) described the recurrent warm-season weather pattern in the region, directing Mistral and Tramontana (NW and N through the Gulf of Lion) winds into the lower layers of the WMB through the Gulf of Lion. This flow has a diurnal pulsation, shifting eastward during the daytime and southward in the late evening and night due to general compensatory subsidence coinciding with sea breezes. Consequently, air masses entering through the Gulf of Lion at night can
be captured by sea breezes along the eastern coast of Iberia, re-entering in the same cycles over several days.

In this context, accumulation by the vertical recirculation of air masses loaded with $O_3$ may occur over several days, particularly during the typical summer absence of significant synoptic advections. Therefore, under persistent anticyclonic conditions, air masses can gradually accumulate $O_3$, leading to progressively higher concentrations in the same areas on subsequent days. These situations typically end with the arrival of a frontal system, which vents out the accumulated $O_3$ and can eventually
transport prefrontal dust-loaded African air masses with different properties leading the cold front (Millán et al., 1997, 2000, 2002; Toll and Baldasano, 2000; Gangoiti et al., 2001; Pérez et al., 2004; Gangoiti et al., 2006; Jiménez et al., 2005; Diéguez at al., 2009, 2014; Millán, 2014; Querol et al., 2017).

In the study area, these conditions can lead to either an open or closed circulation system (Querol et al., 2017 and references therein). Thus, the BMA can act as a significant source of $O_3$ precursors, while downwind areas act as receptor zones. Hence,
we examined the evolution of concentrations of surface $O_3$ not only at AQMSs within the city where the episodes were recorded but also at locations along the mentioned axes and surroundings (Fig. 1c).

## 2.2 Observational data

We employed the following datasets:

-Hourly $O_3$, $NO_2$ and NO concentrations provided by the European Environment Agency (EEA,
https://www.eea.europa.eu/data-and-maps/data/aqereporting-9) for all the AQMSs with available $O_3$ measurements during the study periods in southern France and NE Spain;

-Hourly meteorological data (temperature, relative humidity, solar radiation and wind speed/direction) from two meteorological stations located in the study area (Fig. 1) retrieved from the Meteorological Office of Catalonia (https://analisi.transparenciacatalunya.cat/en/Medi-Ambient/Dades-meteorol-giques-de-la-XEMA/nzvn-apee);
-High-resolution vertical meteorological data (temperature, relative humidity and wind speed/direction) from radiosoundings conducted in Barcelona at 00 and 12 UTC provided by the Faculty of Physics, University of Barcelona. We also used estimations of the mixing layer height (MLH) at 12 UTC using the simple parcel method (Holzworth et al., 1964); and



-Daily $NO_2$ tropospheric column observations (only for the 2019 episode) obtained from TROPOMI, a high-resolution nadir-viewing satellite sensor aboard the ESA's Sentinel-5 Precursor (Veefkind et al., 2012). Observations at 13:30 local solar time organised as daily gridded data ($5.5 \times 3.5$ km$^2$ resolution) were derived from the offline operational product (Van Geffen et al., 2019) using a script in Google Earth Engine (Gorelick et al., 2017) and a quality assurance of >0.75.

### 2.3 Modelling tools

We conducted high-resolution backward and forward trajectory simulations using the mesoscale Regional Atmospheric Modelling System (RAMS) and the HYbrid PArticle Concentration and Transport (HYPACT) model. We used three nested square domains, centred in Barcelona, with grid resolutions of 3, 12 and 48 km, extending up to 190, 1000 and 2600 km around the city, respectively. Backward trajectories ending at less than 400 m above ground level (AGL) in the city were used to identify upwind sources of precursors and $O_3$. Tracer particles emitted from a selection of these upwind sources (cities) were then used to evaluate the eventual convergence of forward trajectories into Barcelona, using a similar methodology as in Gangoiti et al. (2001, 2002, 2006, 2011) and more recently in't Veld et al. (2021). Because we used constant emission rates from all the city sources, this tool could not evaluate the real contribution of each source but the efficiency of the atmospheric transport into the planetary boundary layer (PBL) of Barcelona. This was at an hourly resolution during the pollution episode. The methodology is useful for identifying possible location of sources and the potential of convergence from different source regions. The PBL height and ground elevation influence the amount of particles reaching the receptor sites and the distribution of confluent sources. Receptor sites considered in Barcelona were Ciutadella and Palau Reial, located at nearly sea level, as well as Fabra, located 400 metres above sea level. These locations align with the corresponding AQMSs with the same names (Fig. 1).

The PBL height was retrieved from the ERA5 reanalysis from the European Centre for Medium-range Weather Forecasts (ECMWF), https://cds.climate.copernicus.eu/cdsapp#!/dataset/reanalysis-era5-single-levels?tab=form. Data used were hourly ERA5 reanalysis at midday in the nearest grid point to the corresponding coordinates of the Palau Reial AQMS.

We also conducted photochemistry and dispersion simulations using the Comprehensive Air Quality Model with extensions (CAMx, version 6.5, Ramboll Environment and Health, 2018). For a detailed description of the model, see Torre-Pascual et al. (2023), which analysed a 2018 high $O_3$ episode in the Bay of Biscay a few days prior to the 2018 Barcelona episode assessed here. In our study, we used the same general model configurations described in the reference but with higher vertical resolution, 44 levels up to 6000 m AGL, as shown in Section S4. Emissions were simulated using MEGAN 3.0 to account for biogenic emissions and EDGAR v4.3.2 for anthropogenic emissions. The MEGAN model incorporated recently updated Spanish land use and vegetation map databases sourced from the National Forest Inventory, as demonstrated in Torre-Pascual et al., (2021). We utilised version 4.3.2 of the Emission Database for Global Atmospheric Research (EDGAR) global anthropogenic emission



inventory (Crippa et al., 2018), published in December 2017. This inventory includes anthropogenic emissions from the European and African continents within our study area. We focused on analysing the following key compounds for tropospheric $O_3$ pollution episodes: CO, $NH_3$, VOCs, NOx, $SO_2$ and $CH_4$.

The results of photochemical models in air pollution studies typically present pollutant concentrations and wind fields at the surface level. To enhance the visualisation and interpretation of the dynamics driving $O_3$ episodes, Torre-Pascual et al. (2023) recommended expanding the analyses to higher atmospheric levels, particularly in complex terrains, such as the Spanish territory, due to the multiple processes involved. In this study, we present the results of integrated $O_3$ concentrations and average wind fields at up to 500 m AGL, surface and upper-level $O_3$ concentrations and wind fields, and vertical cross sections along the Besòs-axis to illustrate recirculations, fumigations and subsidence processes.

## 3. Results and discussion

### 3.1 Extreme episodes in Barcelona

Table 1 summarises the only three episodes when Europe's IT has been exceeded in Barcelona since (at least) 2000. Notably, during the 2019 episode, concentrations approached Europe's hourly alert threshold (AT, $O_3$ concentration >240 µg·m$^{-3}$).

**Table 1. General information about the episodes with Europe's Information Threshold (IT) exceedances. Types of air quality monitoring stations (AQMSs) include UB−urban background, SB−suburban background and UT−urban traffic.**

| Date of IT exceedance (dd/mm/yyyy) | Day of the week | Start time (h UTC) | Number of hours above the IT | Name of AQMS (Type) | $O_3$ hourly max. concentration (µg·m$^{-3}$) | Declared heatwave |
|---|---|---|---|---|---|---|
| 06/06/2015 | Saturday | 13 | 1 | Ciutadella (UB) | 206 | no |
| 04/08/2018 | Saturday | 14 | 1 | Fabra (SB) | 190 | yes |
| | | 16 | 3 | Fabra (SB) | 197 | |
| | | 16 | 2 | Gràcia (UT) | 197 | |
| | | 18 | 1 | V. Hebron (UB) | 187 | |
| 05/08/2018 | Sunday | 13 | 2 | Ciutadella (UB) | 190 | |
| | | 15 | 1 | Fabra (SB) | 182 | |
| 29/06/2019 | Saturday | 14 | 2 | Ciutadella (UB) | 190 | yes |
| | | 14 | 2 | Eixample (UT) | 211 | |
| | | 14 | 9 | Fabra (SB) | 236 | |
| | | 15 | 1 | P. Reial (UB) | 229 | |
| | | 16 | 1 | V. Hebron (UB) | 196 | |
| | | 18 | 1 | V. Hebron (UB) | 184 | |





Below, we present a detailed analysis of the three episodes individually (subsections 3.2, 3.3 and 3.4). Within each subsection, we present and discuss results related to the meteorological context, observations and simulations. Annexes are included for each episode (Sections S1 to S3 in the supplemental information) to organise the extensive supplementary material.

### 3.2 2015 episode (6 June)

#### 3.2.1 Meteorological context

*Synoptic situation*

From 1 to 5 June 2015, a low-pressure system was located NW of the British Isles (observed from 3 June in Fig. 2), while a high-pressure ridge moved from SW to NE across the IP towards central Europe. This situation favoured persistent E winds in the WMB, which turned southeast along the Catalan coast and then south with sea breezes. The ERA-INTERIM reanalysis indicated a 50 ppb $O_3$ accumulation at low levels (1000–925 hPa) over the Catalan coast, possibly due to the mentioned persistent E winds from various origins, including Italy and the Balearic Islands. In this scenario, the usual diurnal Tramontana cycle (Gangoiti et al., 2001) was absent, and in the interior of the IP, there might have been the uncoupling of nocturnal drainage flows with valley-down or seaward winds and diurnal couplings with the combined flows of coastal breezes and synoptic E winds.

On 6 June, there was a change in surface pressures in the Bay of Biscay and southern France, with higher pressures than in the Mediterranean (Fig. 2). As a result, Tramontana (N) winds developed and persisted until 9 June, at which time there was a change in the synoptic conditions (not shown).

*Local observations*

Notably, the maximum temperatures (Fig. S1.2d) during this episode in the Barcelona area were 3–4°C higher than the usual June temperatures (Observatori Fabra, 2022), despite it not being a declared heatwave as with the other episodes.

The radiosoundings conducted in Barcelona (Fig. S1.1) showed persistent E winds above 1500–2000 m ASL throughout the period, while land–sea breezes and Tramontana cycles dominated in the lower layers. Before 6 June, lower atmosphere winds followed a daily pattern of W–SW at night, shifting to the S during the day (Fig. S1.1), resulting in a net inland transport towards the N due to coastal sea breeze regimes. The change in synoptic conditions on 6 June caused a noticeable change in the lower atmosphere winds, with an abrupt shift to the E in the early morning. This was accompanied by a temperature increase and relative humidity decrease (Fig. S1.2a–b), likely related to a brief period of vertical coupling with dry easterlies from the free atmosphere (Fig. S1.1). Then, a progressive increase in humidity, coinciding with the wind backing to the S (Fig. S1.2b), marked the onset of the sea breeze. Subsequently, a surface temperature inversion developed at 200 m during the following night, leading to calm conditions in a saturated air layer completely decoupled from the easterlies above.






**Figure 2. Episode 2015. For each day, (top) Climate Forecast System Reanalysis for the 500 hPa geopotential heights and mean sea level pressure (hPa) at 00 UTC (source: www.wetterzentrale.de), (bottom) ERA-INTERIM (ECMWF) reanalysis (0.75° resolution) of O₃ concentration (ppb, contour-lines), and relative humidity (shaded colours) and wind field on the 1000–925 hPa level.**





### 3.2.2 Surface O₃ concentrations

From 3 to 5 June, diurnal $O_3$ concentrations in the BMA (Fig. 3) were relatively low to moderate (up to 60–113 $\mu g \cdot m^{-3}$), displaying a nearly square-shaped daily cycle observed in low-altitude coastal locations in the WMB (e.g. Millan et al., 2000). This pattern showed peak concentrations during the central diurnal hours and the lowest levels at night (Fig. 3b–c). Traffic stations (Eixample, Montcada or Granollers) had minimal concentrations due to $O_3$ titration from local emissions (e.g., Solberg et al., 2005). On 4 and 5 June, the aforementioned persistent moderate S winds in the lower 1000 m of the atmosphere led to

significantly high $O_3$ concentrations downwind of the BMA. This was evident in AQMSs, such as Berga (at the end of the Llobregat-axis) and stations in the Vic Plain (at the end of the Besòs-axis), where the IT was either exceeded or nearly reached (Fig. 3a and c).

The $O_3$ daily cycles along the Besòs-axis (Fig. 3c) and the aforementioned combined upslope winds and sea breezes suggest the usual dynamics of summer $O_3$ episodes downwind (northward) of Barcelona (e.g., Querol et al., 2017) in the days before

the episode. This recurring pattern, termed "Besòs-axis dynamics", is characterised by the gradual increase in diurnal $O_3$ concentrations from Barcelona to the Vic Plain (see the $O_3$ gradients highlighted in Fig. 3c) and the early-afternoon $O_3$ peaks that occur at later times based on their distance from Barcelona, illustrating the northwards path of the BMA plume loaded with new $O_3$ contributions.

The rural background station in Montseny (~700 m ASL) showed minimal daily $O_3$ variation, with concentrations remaining

constant at approximately 120 $\mu g \cdot m^{-3}$ (Fig. 3c). This is due to the limited nighttime consumption, owing to its remote location away from $O_3$-consuming emissions and above the stable nocturnal boundary layer, ensuring a continuous supply of $O_3$ from reservoir layers (e.g., Millán et al., 2000; Millán et al., 2002; Chevalier et al., 2007). A detailed discussion of multiple typified $O_3$ cycles in an air basin, with similar $O_3$ phenomenology to our study area, can be found in Millán et al. (2000).

On Saturday, 6 June (episode in Barcelona), only one station exceeded the IT, but most sites in the city, including the traffic

type, recorded very high $O_3$ concentrations (up to 179 $\mu g \cdot m^{-3}$) and a considerable concentration peak at 12–13 UTC (Fig. 3b). This coincided with the shift in surface winds from the NE to SE during the afternoon (Fig .S1.2a). Stations along the Besòs-axis, farther from Barcelona, recorded significantly lower diurnal $O_3$ concentrations compared to the preceding days (<140 $\mu g \cdot m^{-3}$), with constant levels in the Vic Plain, lacking the usual early-afternoon $O_3$ peak (Fig. 3c). The development of a weaker breeze, as mentioned earlier, along with reduced weekend emissions of precursors from the BMA (see NOx

concentrations in Fig. S1.3), likely contributed to this pattern. This behaviour is common during summer weekends, with the Vic Plain typically exhibiting lower $O_3$ levels than on weekdays (e.g., Massagué et al., 2019).





**Figure 3. Episode 2015. (a) Spatial distribution of maximum hourly O₃ concentrations. Daily O₃ cycles at the surface stations: (b)**
**Barcelona city (no data from Fabra), (c) along the Besòs-axis, ordered with increasing distance to Barcelona: Ciutadella, Montcada,**
**Granollers, Montseny, Tona and Vic. Horizontal red lines represent Information and Alert Thresholds of the EU's Directive (IT**
**and AT).**

In the subsequent days, peak O₃ concentrations in the city decreased but remained significantly higher than before the episode
(Fig. 3c), possibly due to the aforementioned O₃ vertical recirculations and the stagnation of air masses over the Barcelona
area. This situation contributed to the exceedances of the IT at two relatively close coastal stations on 7 and 8 June (Fig. 3a).
By 9 June, O₃ concentrations in the city returned to pre-episode levels (not shown) because of changing meteorological
conditions.



### 3.2.3 Modelling output

***Trajectory analyses results***

The RAMS/HYPACT analyses (Fig. S1.4) show two groups of arrival trajectories to Barcelona (Ciutadella). First, trajectories on 6 June crossed the WMB with E–SE winds from 3 to 5 June, arriving in Barcelona from the E and S, following the Mediterranean gyre (Fig. S1.4a–e). Second, on 7 June, trajectories passed through southern France from 5 and 6 June with W winds, and then transitioned through the Gulf of Lion with the Tramontana winds (Fig. S1.4c–e).

       The temporal evolution of tracer particles (Fig. 4a), emitted in forward trajectories from 18 cities (coloured dots in Fig. 4b)
reaching Barcelona's PBL, indicates the highest impacts originated from multiple sources, including Marseille, Toulouse, Tarragona, Palma (de Mallorca), Cagliari, Montpellier, and Barcelona. This was different from the previous days, where the origin was solely Mediterranean: from the E (Palma, Cagliari) and S (Tarragona).

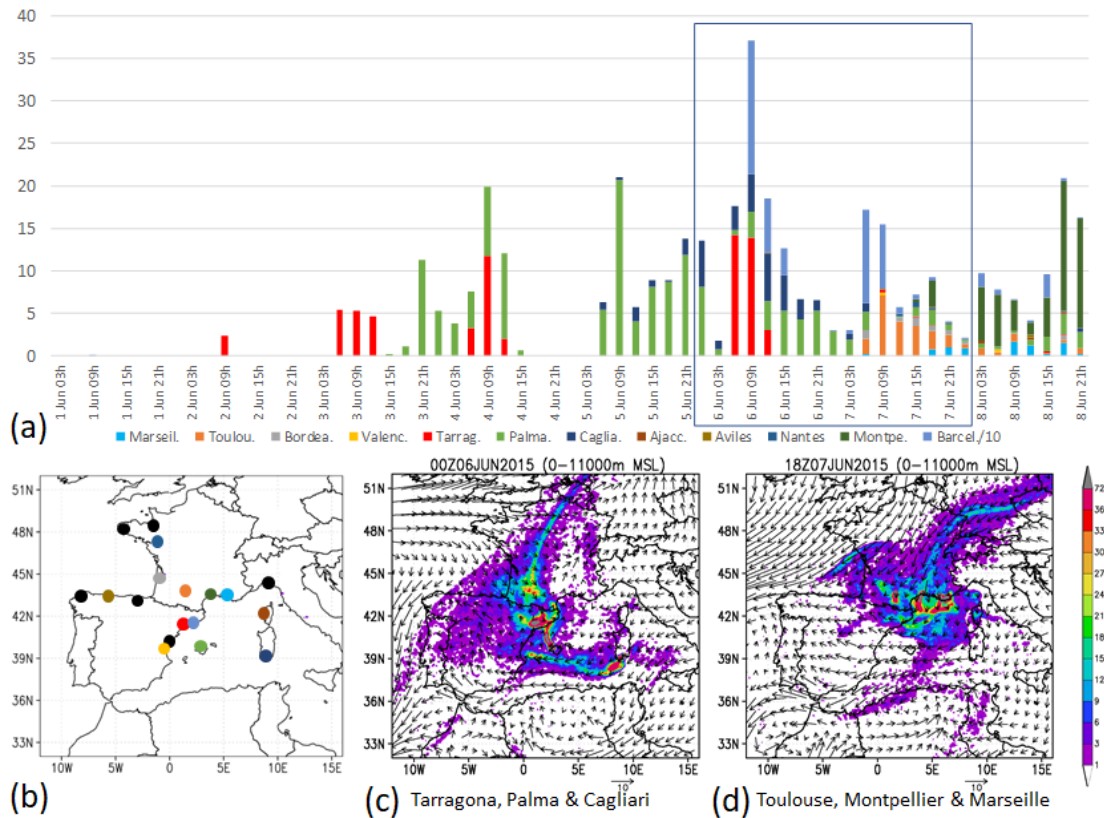

       **Figure 4. Episode 2015. (a) Temporal evolution of impacts (number of tracer particles) emitted from cities shown in the map (b) and**
**reaching Barcelona PBL (Ciutadella) in forward trajectories. Emission point "Barcelona" impacts were divided by 10 due to proximity to the receptor site. Cities not represented in (a) have a negligible contribution, shown as black dots on the map. Sources**





**in (a) use the same colour code as in (b) map. (c–d) Burden of tracer particles (0–11,000 m) impacting Barcelona during the weekend from three selected Mediterranean cities with the main impact on 6 June (c) from the E and S, and (d) three more sources in southern France with the main impact on 7 June from the N.**

Figure S1.5 depicts a Barcelona-originating plume dispersion simulation with northward transport before 6 June and no circulation of coastal emissions over the WMB. Starting on 6 June with Tramontana winds, a land–sea recirculation developed along the Barcelona coast. The plume underwent a complete rotation, with Barcelona at the centre of a convergence zone. Barcelona received pollution from the Gulf of Lion and from the E and SE areas, transported by sea breezes and the Mediterranean gyre. These inputs merged with the local emissions from the E Spanish coast.

The key finding of these analyses is the convergence of a set of multiple sources on the day of the episode, located to the NW of the city (S France), to the E (other Mediterranean cities), to the S (Tarragona) and in Barcelona.

*Photochemical model results*

Outcomes from the photochemical simulation from 5 June are shown in Figs. 5 and S1.6. The model replicates intense sea breezes along the Catalan coast, aligning with local observations. These breezes transported the BMA plume northeastward,
reaching nearly the Pyrenees (Fig. S1.6). The vertical section in the same figure also shows $O_3$ accumulation northward of the Pyrenees (circle on 5 June at 16 UTC in Fig. S1.6). The presence of a high pressure over the Balearic Islands suggests subsidence of upper layers over the sea. In the afternoon, stagnant high-altitude air masses over Barcelona contributed to $O_3$ accumulation (Fig. S1.6), consistent with the ERA-INTERIM results. Notably, Fig. 5 indicates a localised transport of $O_3$ from Barcelona/Girona towards the Gulf of Lion.

On the early morning of 6 June (episode day in Barcelona), this contribution added to $O_3$ from the Gulf of Lion and returned over the Catalan coast, coinciding with the onset of the Tramontana wind (Fig. 5). This is consistent with the back trajectories analyses. At this point, there was no longer $O_3$ accumulation northward of the Pyrenees (circle on 6 June at 00 UTC in Fig. S1.6). On 6 June, the impact of the transported air masses from the previous day (Fig. 5) and the vertically recirculated $O_3$ (Fig. S1.6) in the city were observed. Simulated breezes were not intense, consistent with local observations, and this partially
disrupted the Besòs-axis dynamics and facilitated the increase in $O_3$ in the city. Additionally, the potential impact of the weekend effect should be considered.

Simulations show vertical recirculations of $O_3$ towards the sea on 7 and 8 June and low $O_3$ levels in the Gulf of Lion. The Tramontana showed limited $O_3$ transport (Figs. 5 and S1.6). The $O_3$ accumulated aloft and over the sea surface (mostly from the preceding days) shifted southwestward, affecting other areas (Figs. 5 and S1.6). This is consistent with the observed $O_3$
surface concentrations, which decreased in Barcelona but still remained high in other coastal cities where the IT was exceeded, e.g. Tarragona. On 8 June, a similar transport pattern was observed over the WMB with a slight decrease in the $O_3$ concentration, probably partially due to the restoration of weekday emissions of $O_3$-consuming compounds. This pattern



persisted until 9 and 10 June (not shown), at which time the Tramontana wind ceased, and E winds transported surface $O_3$ westward, inhibiting the accumulation processes of the previous days.

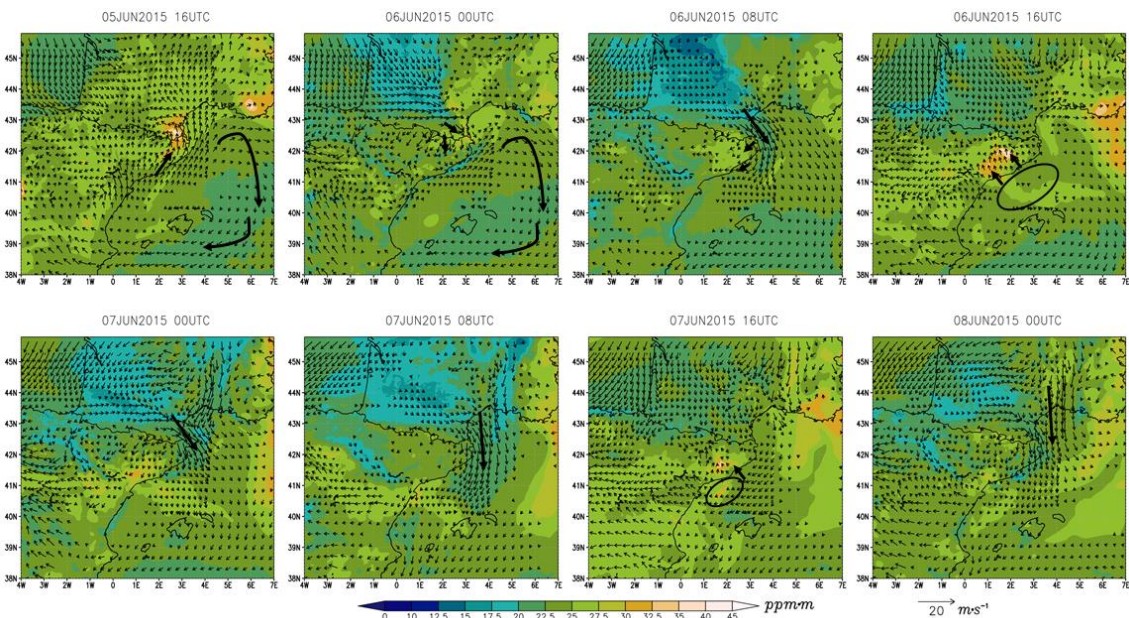

**Figure 5. Episode 2015. Simulated $O_3$ concentrations (colour scale, in ppm·m) integrated at 0–500 m above ground level (AGL) and average wind fields (vectors) between 0 and 500 m AGL. Wind speeds <2 m·s⁻¹ are not represented.**

### 3.3 2018 Episode 2018 (4 and 5 August)

#### 3.3.1 Meteorological context

*Synoptic situation*

From 1 to 3 August 2018, NE winds (1000–900 hPa) persisted over the W part of the European continent, from the Jutland peninsula to northern Iberia (Fig. 6), giving rise to Tramontana winds in the study area. This flow, associated with high-pressure systems over Scandinavia, extended through a broad NE–SW ridge to Iberia and North Africa. On the afternoon of 3 August, the anticyclone retreated from Jutland, strengthening W of Ireland on 4 and 5 August (episode days). Consequently, the NE winds rotated N along the Atlantic European coast, while the Tramontana persisted, driven by the pressure difference between southern France and the Mediterranean. The ERA-INTERIM reanalysis shows E winds in the Gibraltar Strait from 1 to 5 August, along with the Tramontana winds, following the continuity of the airflow in the marine boundary layer of the WMB (as in in't Veld et al., 2021). This airflow, which was more evident at nighttime, moved into the relative lower pressure in the Gulf of Cadiz, following the eastern coast of Iberia. On 6 August, as the surface high pressures weakened in the Bay of Biscay and western France and the anticyclone shifted towards western Portugal, meteorological conditions changed,




preceding the approximation of an Atlantic front through the W of the Iberian Peninsula. This marked the conclusion of the Tramontana period and the decay of the episode.

**Figure 6. Episode 2018.** For each day, (top) Climate Forecast System Reanalysis for the 500 hPa geopotential heights and mean sea level pressure (hPa) at 00 UTC (source: www.wetterzentrale.de), (bottom) ERA-INTERIM (ECMWF) reanalysis (0.75° resolution) of O₃ concentration (ppb, contour-lines), relative humidity (shaded colours) and wind field on the 950–850 hPa level.




*Local observations*

The most intense heat episode of 2018 occurred from the first few days of August until 8 August, with coastal regions, including Barcelona, experiencing exceptionally high temperatures, and some set record highs. The peak temperatures during the episode days (Fig. S2.2b) in the city were conditioned by a light N–NE flow. This limited the tempering effect of the southerly sea breezes, which weakened during the episode, as discussed below (Meteocat, 2022).

        The radiosoundings conducted in Barcelona (Fig. S2.1) show well-developed sea breezes from 1 to 3 August, also observed
at the Barcelona and Parets stations (Fig. S2.2a and c). In the upper levels of the atmosphere, the wind shifted to a N pattern on 2 August at 12 UTC, which probably resulted in the entry of the upper-level polluted air masses documented by Torre-Pascual et al. (2023). On 3 and 4 August, MLHs were high between 2000 and 3000 m AGL in the city (data from 5 August are not available). The wind pattern observed at the surface on 4 and 5 August was different from the previous days, likely suggesting a weaker sea breeze development during that period (Fig. S2.2a and c), also suggested by the model simulations
(see below).

        On 6 and 7 August, the previously mentioned meteorological change became evident on surface measurements, with a drop in maximum temperatures (–6°C), an increase in relative humidity (+30–35%), and weaker solar radiation compared to the episode days (Fig. S2.2a).

### 3.3.2 Surface $O_3$ concentrations

In early August 2018, $O_3$ concentrations were high on a regional scale. For instance, AQMSs in S France reached extreme $O_3$ levels, with several AT exceedances northward from Marseille (Fig. 7a), and the Atlantic coast of the IP showed abnormally high concentrations, according to Torre-Pascual et al. (2023). Background concentrations in the study area from 1 to 3 August were high, as background stations in Barcelona (Ciutadella, Fabra and V. Hebron) recorded concentrations of up to 140 μg·m$^{-3}$ (Fig. 7b). The Besòs-axis dynamics appear to be well established, causing very high diurnal $O_3$ concentrations, (up to 214
μg·m$^{-3}$ in the Vic Plain) and multiple exceedances of the IT (Fig. 7a and c). On 3 August, peak $O_3$ concentrations decreased in the Vic Plain but remained close to the IT.





**Figure 7.** Episode 2018. (a) Spatial distribution of maximum hourly O₃ concentrations. Daily O₃ cycles at the surface stations: (b) Barcelona (no data from P. Reial), (c) along the Besòs-axis, ordered with increasing distance to Barcelona: Ciutadella, Montcada, Granollers, Montseny, Tona and Vic. Horizontal red lines represent the Information and Alert Thresholds of the EU's Directive (IT and AT).

During the episode in Barcelona (4 and 5 August), multiple AQMSs in the city and the BMA recorded very high O₃ concentrations, reaching up to 197 µg·m⁻³ (Fig. 7a). At the suburban Fabra station in Barcelona, exceptionally high concentrations were observed during the night of 4 to 5 August. Along the Besòs-axis, although the O₃ concentrations remained high and close to the IT, they significantly decreased compared to 1 and 2 August, and no O₃ gradient was observed (Fig. 7c), suggesting the disruption of the Besòs-axis dynamics. This resembled the 2015 episode and could have been partially attributed





to the aforementioned reduced development of breezes during the episode days and to the lower weekend precursor emissions in the BMA, as indicated by the low NOx concentrations (Fig. S2.3).

The approximation of the Atlantic front on 6 August (Monday) caused the episode in Barcelona to decay. Diurnal $O_3$
concentrations dropped to 80–140 $\mu g \cdot m^{-3}$ (Fig. 7b), especially at low-altitude stations, likely partially due to the restoration of local emissions of $O_3$-consuming compounds typical of workdays. The Besòs-axis dynamics seemed to re-establish, displaying a marked gradient of $O_3$ concentrations in AQMSs farther N and concentrations of up to 206 $\mu g \cdot m^{-3}$ in the Vic Plain Fig. 7c).

### 3.3.3 Modelling output

*Trajectory analyses results*

The RAMS/HYPACT analyses (Fig. S2.4) show the arrival of tracer trajectories from central Europe to Barcelona (Fabra) on 4 and 5 August (episode days), crossing France. Air masses entered the WMB, following the Tramontana winds associated with relatively high pressures over Europe. Additionally, daytime sea breeze circulations along the E coast of Iberia generated southerly flows, potentially recirculating precursors and $O_3$ and adding new sources, including Tarragona and a short impact from Castellón.

Figure 8b shows the temporal evolution of tracer particles emitted in forward trajectories from a selection of 14 selected cities reaching the Barcelona PBL (Ciutadella). The impact of Toulouse, representing the arrival of precursors from S France, was characteristic throughout the Tramontana period. The influence from Tarragona occurred nearly every day but was not consistent. During the episode (marked with a square in Fig. 8b), a convergence of multiple sources from various regions occurred, resulting in higher overall concentrations, similar to the 2015 episode. The main sources on the episode days were
from central Europe, S France and the SE Iberian coast (Fig. 8c–d).



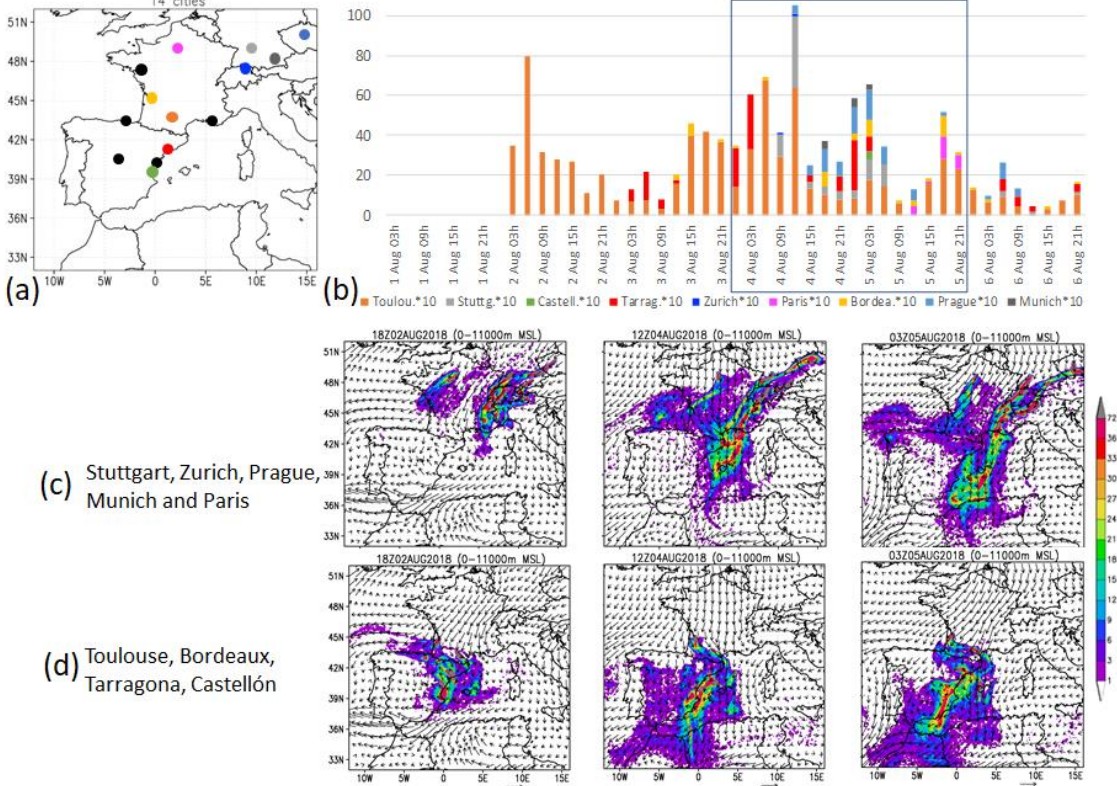

**Figure 8. Episode 2018. (b) Temporal evolution of impacts (number of tracer particles) emitted from cities shown in (a) reaching Barcelona PBL (Ciutadella) in forward trajectories. Cities not represented in the bar chart had a negligible contribution, shown as black dots on the map. (c) Burden of tracer particles (0−11,000 m) impacting Barcelona emitted from Stuttgart, Zurich, Prague, Munich and Paris and (d) Toulouse, Bordeaux, Tarragona and Castellón.**

*Photochemical model results*

The results from the photochemical simulation are shown in Figs. 9 and S2.5. The model reproduces the presence of Tramontana and accumulation of $O_3$ at mid-tropospheric levels across most of Iberia on 2 and 3 August (see Torre-Pascual et al., 2023). Transport of polluted air masses entering the western Mediterranean from the Gulf of Lion can be observed (Fig. 9), consistent with the trajectory results. For these days, simulations reproduced the breezes and return flows on the E coast (Figs. 9 and S2.5), causing an increase in $O_3$ over the sea in front of Barcelona, as also shown in the ERA-INTERIM reanalysis (Fig. 6). These mechanisms caused an accumulation of $O_3$ at the surface, resulting both from transport over the sea and local production.

The results indicate that on both 4 and 5 August (episode days), Barcelona was impacted by the arrival of polluted air masses from Europe through the Gulf of Lion (Figs. 9 and S2.5), consistent with the trajectory analyses. Furthermore, $O_3$ contributions from the S (e.g., from Tarragona) on 4 August (Fig. 9) and the accumulation of $O_3$ over the sea from the vertical recirculations



on the previous days affected the city. Return flows continued mixing upper-level $O_3$ with that situated below, contributing to more accumulation at the surface. Additionally, the intensity of the episode was likely influenced again by weakened sea breezes on 4 August (Fig. 9), as also suggested by local observations, resulting in less inland penetration of polluted air masses than on previous days. This pattern is similar to the 2015 episode. Similarly, as in the 2015 event, the weekend effect should also be considered, which likely contributed to reduced $O_3$ titration and ozonolysis.

On 6 August (Monday), $O_3$ concentrations decreased owing to meteorological changes and the recovery of $O_3$-consuming emissions. This decay of the episode is reflected in the simulations with the end of the Tramontana winds and the absence of discernible $O_3$ contributions from Europe (6 August at 00 UTC in Fig. 9).

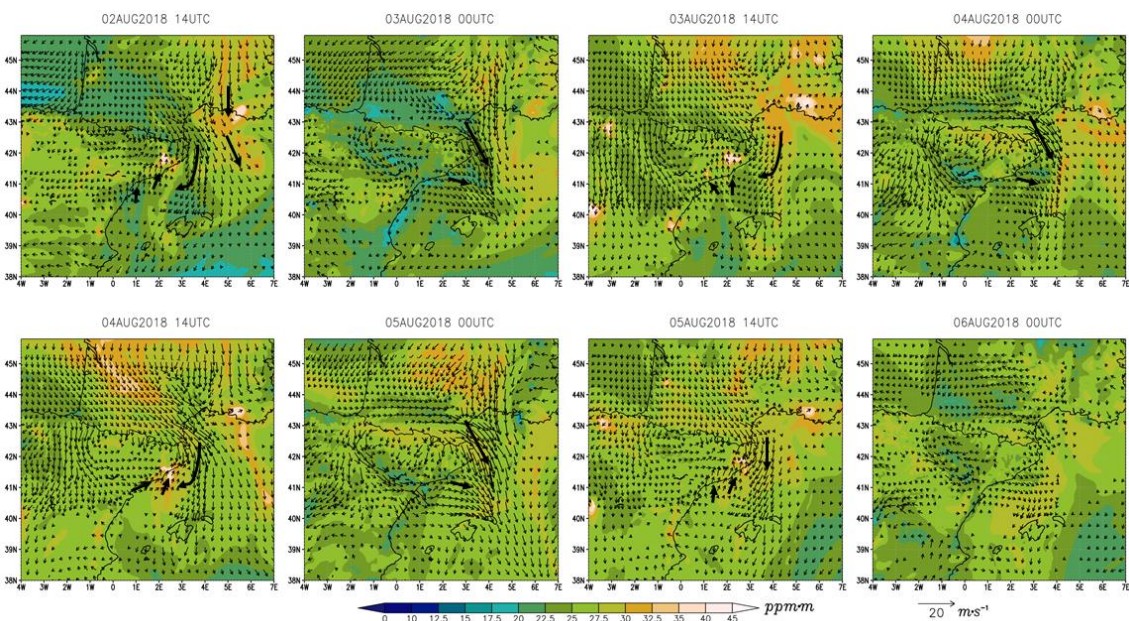

**Figure 9. Episode 2018. Simulated $O_3$ concentrations (colour scale, in ppm·m) integrated at 0–500 m above ground level (AGL) and average wind fields (vectors) between 0 and 500 m AGL. Wind speeds <2 m·s⁻¹ are not represented.**

**3.4 2019 episode (29 June)**

**3.4.1 Meteorological context**

*Synoptic situation*

From 20 to 25 June 2019, a low-pressure centre detached from the main westerly current and moved to the NW of Iberia, weakening at the surface during its trajectory (not shown). By 25 June, it persisted as an upper-level low, isolated from the polar front. Simultaneously, a high-pressure ridge developed over the WMB after its associated African warm S winds advanced ahead of the low-pressure centre in the Atlantic sector, inducing a heatwave event in E Iberia, including in Barcelona



(Fig. 10). This ridge was particularly well defined on 25 and 26 June, as mid-tropospheric S winds were observed over the Barcelona sounding site (Fig. S3.1). In the lower layers, surface low pressures on the coasts of Portugal and the Bay of Biscay, associated with the described upper-level low and relatively higher surface pressures on the WMB, inhibited the Tramontana winds, suppressing the diurnal pulsation in the WMB (Gangoiti et al., 2001). This situation led to persistent easterly circulation from the central Mediterranean to the E Iberian coast until 27 June (Fig. 10), possibly contributing to the accumulation of $O_3$

(55–60 ppb), as also shown in the ERA-INTERIM reanalysis (see below). This was likely due to the transport of $O_3$ precursors from Italy and North Africa.

On 28 June, the Atlantic low-pressure centre started displacing northwards to integrate again with a polar front trough NW of Ireland on 30 June. During this transition, the Tramontana flow was activated, and W–NW winds prevailed up to a height of 1000–1500 m (Fig. S3.1, from 28 June at 12 UTC to 29 June), which caused a drastic decrease in humidity at surface stations

and the maximum temperature during the period, even at nighttime (see next section). The sea breeze regime and inland convergence (SW, in Fig. S3.1) showed a very shallow (250 m) layer on 28 June at 12 UTC, influenced by a drier and warmer Tramontana flow. These westerlies, associated with the Tramontana, continued blowing during the following day. This situation favoured the development of return flows after the onset of the coastal easterlies and southeasterlies during the sea breeze regime on 29 June, with subsequent pollution accumulation on the coast, as discussed in Section 3.4.3 below. Notably,

the synoptic situation during the build-up and peak of this episode was similar to the previously described 2015 episode. On 30 June, the situation changed with the onset of more intense southerlies above 1000 m in height (Fig. S3.1), preventing the formation of sea breeze return flows and favouring venting conditions.

*Local observations*

The persistent easterlies before the episode were also observed at the surface with steady and relatively intense NE–E winds

(5–9 m·s⁻¹) on 25 and 26 June (Fig. S3.2), which combined with the southerlies above 1000 m (Fig. S3.1) likely favoured the dispersion of pollutants to the NW. The synoptic situation described above caused Catalonia to experience an intense heatwave from 27 to 30 June (Meteocat, 2022), peaking in maximum temperatures from 28 to 29 June. In Barcelona, temperatures increased progressively to over 36°C on 28 June (Fig. S3.2d). During the night before the episode, abnormally high temperatures persisted, possibly due to dry westerlies observed in the soundings (29 June at 00 UTC in Fig. S3.1), with the

absolute minimum relative humidity. We interpret this as a Foehn episode of the Tramontana winds at the lee of the mountain ranges, running parallel to the coast to the W and NW of the city. On 30 June, surface conditions shifted, with the maximum temperatures decreasing by 5°C and the relative humidity peaking at 90%. This coincided with the onset of the marine easterlies at lower levels (29 and 30 June) and the upper-level southerlies on 30 June (Fig. S3.1).





**Figure 10. Episode 2019. For each day, (top) Climate Forecast System Reanalysis for the 500 hPa geopotential heights and the mean sea level pressure (hPa) at 00 UTC (source: www.wetterzentrale.de), (bottom) ERA-INTERIM (ECMWF) reanalysis (0.75° resolution) of O$_3$ concentration (ppb, contour lines), relative humidity (shaded colours) and wind field at 950–850 hPa.**

### 3.4.2 Surface O$_3$ concentrations

On 25 and 26 June, O$_3$ levels in the study area were moderate for that time of year (Fig. 11a), whereas in Barcelona, concentrations (<110 µg·m$^{-3}$) showed little variation throughout the day (Fig. 11b). This stable pattern is attributed to the aforementioned favourable dispersion conditions, which maintained low NOx levels (Fig. S3.2), thus weakening/eliminating



nocturnal titration. On 27 June, the usual nighttime titration in Barcelona was restored (Fig. 11b), and the Besòs-axis dynamics seemed to develop (Fig. 11c), with $O_3$ concentrations reaching up to 193 µg·m$^{-3}$.

On 28 June, $O_3$ concentrations increased significantly along the Besòs-axis, where the Montseny station registered extreme $O_3$
levels (263 µg·m$^{-3}$). Concentrations approached the AT in the Vic Plain (Fig. 11a). Simultaneously, the AT was exceeded at stations along the NE-axis (Fig. 11a), pointing to one of three possible transport routes of the BMA plume (Diéguez et al., 2009). Jaén et al. (2021), who also studied this 2019 episode, indicated that high NOx levels likely favoured $O_3$ production in these downwind (NOx-limited) areas. Indeed, high $NO_2$ levels in the morning were observed in the city, reaching up to 233 µg·m$^{-3}$ (both at the surface, as depicted in Fig. S3.3, and in the tropospheric column, as shown in Fig. S3.4). These elevated
$NO_2$ levels could partially result from the holiday exodus, as the date coincided with a Friday preceding the vacation for a significant portion of the population. Additionally, emissions from a fire in Ribera d'Ebre (Tarragona) from 27 June onwards (clearly observed by TROPOMI, Fig. S3.4) could have contributed to the unusually high $O_3$ concentrations at the Montseny station (located at altitude). However, after subsequent simulations and analysis of trajectories, we concluded that the fire plume passed over Barcelona at a height above 1500 m ASL on the episode day, well above the PBLH, suggesting no direct
effect on $O_3$ levels in the city.

On Saturday, 29 June (episode in Barcelona), extreme $O_3$ concentrations were recorded in the city, with levels reaching up to 236 µg·m$^{-3}$ and nearly all the active stations exceeding the IT (Fig. 11a). Interestingly, $O_3$ concentrations in Barcelona, especially those recorded in Fabra, exhibited a relative maximum at 10 UTC (time of low photochemical production), followed by the absolute peak at approximately 15 UTC, aligning with the typical time of maximum $O_3$ concentrations.

In the SW BMA areas, several stations exceeded the IT, with one station exceeding the AT (Gavà, Fig. 11a). Jaén et al. (2021) attributed this pattern to a shift in the wind conditions, diverting the BMA plume (comprising recirculated and fresh emissions) westward instead of northward, which is consistent with the observed diurnal light surface winds from the E quadrant (Fig. S3.2a) and the model results (see below). Consistent with this, the $O_3$ daily cycles suggest an alternate BMA plume route (Diéguez et al., 2009) along the Llobregat-axis instead of the Besòs-axis (Fig. 11c and d).

In the following days, the episode decayed due to the changing meteorological conditions. This is evident from the considerable decrease in $O_3$ concentrations, particularly at the low-altitude stations in Barcelona, with levels similar to pre-episode days (Fig. 11a).





**Figure 11. Episode 2019. (a) Spatial distribution of maximum hourly O₃ concentrations. Daily O₃ cycles at surface stations: (b) Barcelona, (c) along the Besòs-axis, ordered with increasing distance to Barcelona: P. Reial, Montcada, Granollers, Montseny, Tona and Vic (some hourly data are missing in Granollers and Montseny) and (d) the Llobregat-axis. Horizontal red lines represent Information and Alert Thresholds of the EU's Directive (IT and AT).**

### 3.4.3 Modelling output

*Trajectory analyses results*

The RAMS/HYPACT analyses (Fig. S3.5) revealed two groups of arrival trajectories to Barcelona (Fabra) on 29 June. The first group crossed the WMB with the aforementioned easterly winds on 26 and 27 June and came from central Europe with N winds. The second group on 28 and 29 June came from the S, following the coast ("Mediterranean" contributions) with the Mediterranean Gyre circulation, and from the N with the characteristic entry through the Gulf of Lion, sweeping across S France (continental European contributions).

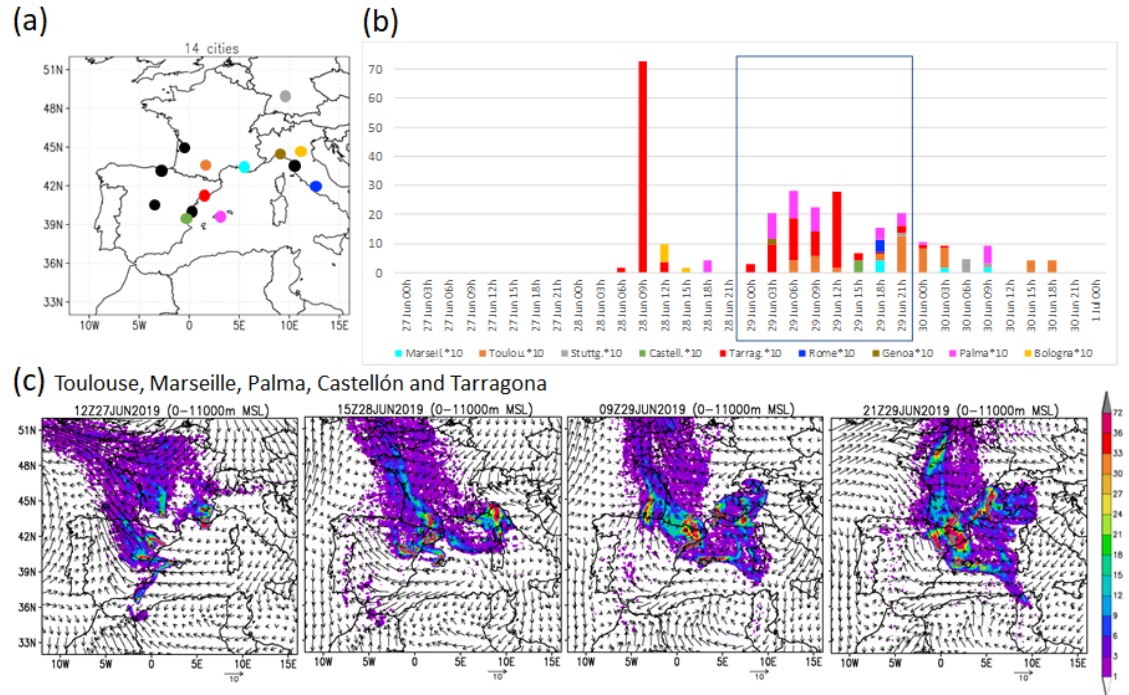

**Figure 12. Episode 2019. (b) Temporal evolution of impacts (number of tracer particles) emitted from cities shown in (a) reaching Barcelona PBL (Fabra). To account for (b) variations in impact intensity between Tarragona and other cities, the concentrations of the remaining cities were multiplied by 10. Cities not represented in the (b) bar chart had negligible contributions, shown as black dots on map (a). (c) Burden of tracer particles (0−11,000 m) impacting Barcelona emitted from five selected cities: Toulouse, Marseille, Palma, Castellón and Tarragona.**





The temporal evolution of tracer particles emitted in forward trajectories from 14 selected cities (Fig. 12b) shows that the Barcelona PBL received impacts from multiple sources, as also observed in the other two episodes. The city lies in a convergence zone of N winds (Tramontana winds carrying pollution from Marseille and Toulouse) and S winds (evening anticyclonic shallow vortex with inland convergence, bringing pollutants from Tarragona, Castellón and Palma). Traces from
Rome, Bologna, Genoa and Stuttgart also appeared, although to a lesser extent (with higher emissions above Barcelona). Figure 12c shows the total particle burden (0–11000 m) emitted from 5 of the 14 cities (Toulouse, Marseille, Palma, Castellón and Tarragona) before the episode (left) and during the episode (right). The joint plume initially moved westward (27 June) within the persistent easterly regime, but on 28 and 29 June, the onset of the Tramontana and the WMB circulations forced its convergence into Barcelona from the S, E and N. Thus, the results again indicate multisource convergence from the main
contributors in France, Spain and Italy.

*Photochemical model results*

The results indicated high $O_3$ concentrations on 27 June along the Besòs-axis, consistent with the observed concentrations (Fig. 13). In the morning of 28 June and throughout the day, the simulated $O_3$ concentrations were slightly higher, shifted towards the NE and located in a relatively calmed area. This was due to the convergence of SW winds and the Tramontana wind,
suggesting $O_3$ accumulation at the night before night. On 28 June, the sea breezes penetrated far less inland than on the previous day. The circulation induced by the Balearic daytime anticyclonic gyre carried the plume from the BMA through the Besòs-axis in a northeastward direction, coinciding with IT exceedances observed at the surface (Fig. 11a). During the night of 28 to 29 June, the Tramontana winds caused the plume to return towards the sea. This plume could be enriched with $O_3$ or precursors arriving from France, as inferred from the increased tropospheric $NO_2$ levels shown in Fig. S3.4. Simulations show high
integrated $O_3$ concentrations over the BMA throughout that night, which with low-wind conditions, resulted in relatively high concentrations the next morning (Fig. 13).

On 29 June (episode day), off the coast of Barcelona over the sea, TROPOMI detected high levels of $NO_2$, consistent with the surface $O_3$ titration suggested by the simulations (upper panels in Fig. S3.6). This was likely caused both by a very light nocturnal sea breeze that carried precursors out to sea and by the high emissions from the holiday exodus mentioned above.
During the morning and midday, with relatively weak sea breezes, the entire $O_3$-polluted air mass that was transported by the Tramontana during the previous day, combined with fresh NOx emissions along the coast, entered into the BMA, leading to the elevated $O_3$ concentrations. Notably, the surface $O_3$ concentrations observed in the city (Fig. 11) displayed two distinct peaks. The first peak was likely caused by fumigation of the $O_3$-polluted air mass before midday (bottom panels in Fig. S3.6), which was present at upper levels in a relatively warm and dry air mass and decoupled from the surface during the nighttime
(Fig. S3.1). These observations highlight the presence of upper-level $O_3$ pollution resulting from the vertical recirculation of pollution initiated on the day before (28 June) due to the onset of the Tramontana northwesterlies and the sea breeze convergence (easterlies) at the coastal strip.





On 30 June, upper-level winds changed to S and SW, also increasing their intensity and persisting for several days (Fig. S3.1). At the surface, sea breezes penetrated further inland, with no significant return flows due to the intensity of the upper-level southerlies. The simulation reproduced this behaviour with its subsequent O$_3$ decrease in the BMA (not shown) and the associated decay of the episode.

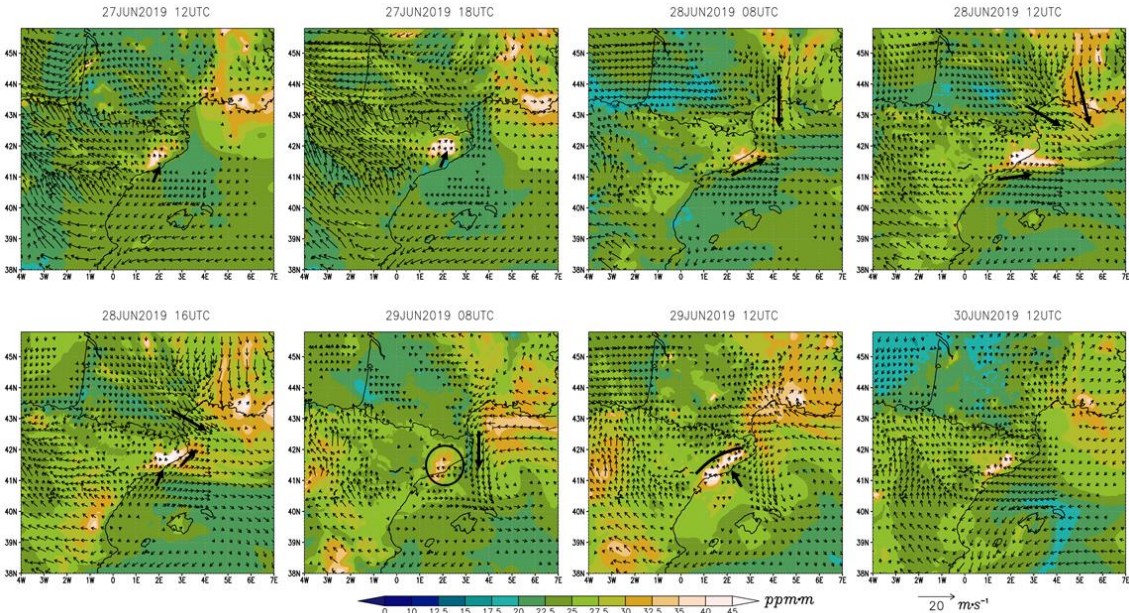

**Figure 13. Episode 2019. Simulated O$_3$ concentrations (colour scale, in ppm·m) integrated at 0–500 m above ground level (AGL) and average wind fields (vectors) between 0 and 500 m AGL. Wind speeds <2 m·s$^{-1}$ are not represented.**

## 4. Summary and conclusions

Since at least the year 2000, Barcelona has experienced three extreme ozone (O$_3$) episodes in which the EU's hourly Information Threshold (180 μg·m$^{-3}$) was exceeded. These episodes occurring in the summers of 2015, 2018 and 2019 were concerning, especially considering that Barcelona is the second most populous city in Spain. This study aimed to analyse their phenomenology comprehensively through experimental data and modelling tools to identify drivers and common patterns, and offer predictive insights.

The build-up of the 2015 episode was characterised by synoptic conditions inhibiting Tramontana (N) winds in the Western Mediterranean Basin (WMB), allowing persistent E winds and subsequent O$_3$ accumulation along the E coast of Iberia. On the episode day, the onset of the Tramontana winds led to the inflow of air masses from S France through the Gulf of Lion and the anticyclonic gyre circulation over the WMB at low levels (Gangoiti et al., 2001). This situation led to (i) the mobilisation of previously accumulated O$_3$ along the coast, (ii) the transport of O$_3$ and its precursors from the coastal areas towards the city,



and (iii) the simultaneous import of pollutants from the Gulf of Lion, converging with the other contributions over Barcelona, including vertically recirculated $O_3$. Additionally, (iv) the weakening of the sea breezes, (v) the weekend effect caused by the lower consumption of $O_3$ from reduced NO and VOC emissions, and (vi) the unusually high temperatures contributed to the exacerbation of $O_3$ levels in the city during the episode.

During the development and peak of the 2018 episode, synoptic conditions favoured the persistence of Tramontana winds in the WMB. The build-up of this episode was characterised by high $O_3$ concentrations at ground level and high altitudes on a regional scale, resulting in initially high background levels in the city. On the day of the episode, this $O_3$ contribution was added to (i) the influx of European polluted air masses, which converged with other polluted air masses from multiple sources, including the E Iberian coast and S France, and (ii) vertically recirculated $O_3$ from the previous days over the sea. Additionally,
(iii) weakened breezes, (iv) the weekend effect, and (v) the exceptionally high temperatures contributed to the increased $O_3$ levels in the city.

The 2019 episode was characterised by similar synoptic conditions, during its build-up and peak to that of the 2015 episode. Again, the initial inhibition of the Tramontana winds led to persistent easterly circulation from the central Mediterranean that contributed to the accumulation of $O_3$ along the coast. Subsequently, the Tramontana was activated, adding more $O_3$ through
(i) the incorporation of new sources of pollutants from S France towards the WMB and (ii) the mobilisation of pollutants previously located/emitted on the SE Iberian coast towards Barcelona, also incorporating coastal emissions due to the characteristic anticyclonic gyre in the WMB. Additionally, as in the other episodes, (iii) the weekend effect and (iv) the abnormally high temperatures contributed to the very high $O_3$ concentrations.

The results indicate a complex interplay of multiple factors that caused the three episodes, also revealing specific shared
commonalities that contributed to their occurrence. These included (i) initial regional accumulation of $O_3$, (ii) the presence of Tramontana winds, which may have resulted in $O_3$ and precursors transporting from France into the WMB (although the mere presence of these winds does not necessarily imply the observed transport to Barcelona or the development of an extreme $O_3$ episode), (iii) the accumulation from vertical recirculation processes the previous days of the episode, including the horizontal circulation of the local plume in the weakened sea breeze regime, (iv) convergence of polluted air masses from multiple sources
to the S (Tarragona), N and NW (Marseille, Toulouse) and, from the E (Mallorca and Sardinia with the continental sources in Italy), (v) calm winds in the upper layers, (vi) the weekend effect, and (vii) abnormally high temperatures.

Thus, combined factors provide valuable insights into predicting the likelihood of episodes in the city, particularly when considering prevalent meteorological conditions. These conditions may include the forecasted stagnation of air masses at higher altitudes, weakened or absent breezes, observed recirculation and accumulation processes in the days leading up to the
event, heatwaves or exceptionally high temperatures, or the presence of Tramontana winds in the WMB. Other conditions,



such as the proximity to holidays or weekends, and other pertinent factors should also be considered. Hence, the comprehensive evaluation of these factors should be imperative for the future of air quality management in this region.

**Author contributions**

JM: Conceptualization; Data curation; Formal analysis; Investigation; Methodology; Software; Validation; Visualization; Writing – original draft; Writing – review & editing

ET-P: Conceptualization; Data curation; Formal analysis; Investigation; Methodology; Software; Validation; Visualization; Writing – original draft; Writing – review & editing

CC: Conceptualization; Data curation; Formal analysis; Investigation; Methodology; Software; Visualization; Writing – review & editing

ME: Conceptualization; Formal analysis; Investigation; Supervision; Methodology; Software; Validation; Visualization; Writing – original draft; Writing – review & editing

AA: Investigation; Funding acquisition; Supervision; Formal analysis; Investigation; Methodology; Software; Validation; Visualization; Writing – original draft; Writing – review & editing

MP: Data curation; Formal analysis; Investigation; Methodology; Software; Visualization; Writing – review & editing

XQ: Conceptualization; Formal analysis; Funding acquisition; Investigation; Supervision; Methodology; Validation; Visualization; Writing – review & editing

GG: Conceptualization; Data curation; Formal analysis; Investigation; Supervision; Methodology; Software; Validation; Visualization; Writing – original draft; Writing – review & editing

**Competing interests**

At least one of the (co-)authors is a member of the editorial board of Atmospheric Chemistry and Physics

**Acknowledgements**

This study is supported by the Spanish Ministry of Ecological Transition and Demographic Challenge, in the framework of the Spanish Ozone Abatement Plan. The IDAEA-CSIC team also acknowledges the support received from the Generalitat de Catalunya (AGAUR 2021 SGR 00447). We would like to thank Yolanda Sola (University of Barcelona, Faculty of Physics) for providing data from the radiosoundings, EEA for the air quality data from European monitoring stations, Meteocat for the meteorological surface data, ESA for remote-sensing $NO_2$ data (TROPOMI), Wetterzentrale.de for the synoptic maps and QGIS for their GIS software.



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
