# Peer review of "Extreme ozone episodes in a major Mediterranean urban area"

_EGUsphere, 2023_

## Author Comment (AC1)

**Responses to referees' comments on egusphere-2023-2449:**

We thank the reviewers for reading carefully our study and providing valuable questions, comments and recommendations, which helped us to greatly improve our work.

In this document, we provide point-to-point responses to every question and comment provided by both referees (Referee #1 and #2). Referees' comments are presented in black, our responses in blue, and, when applicable, the reproductions of the revised texts are displayed in both quotation marks ("") and *italics*. References to line numbers in the text refer to the revised version.

Additionally to the modifications resulting from the referees' feedback, we have made other minor adjustments, corrections and enhancements throughout the manuscript to enhance its overall coherence and flow.

**Referee #1**
**https://doi.org/10.5194/egusphere-2023-2449-RC1**

This paper analyzed three O3 episodes that occurred in Barcelona, Spain during the summers of 2015, 2018 and 2019. During the three episodes, the EU O3 threshold was exceeded. The paper aims at investigating the complex factors affecting the episodes by using observational data as well as simulations. Although the topic is interesting and the 3 episodes have been described in detail with comprehensive data, the paper is not well structured and the main scientific research findings are not clearly presented.

We greatly appreciate the constructive feedback and have made efforts to address all the issues raised by the reviewer.

1. Abstract, there are too many factors presented (7 in total), but there is a lacking of the logistics among these factors. How do these factors influence the 3 extreme ozone episodes? What are commonalities and differences of these factors among the 3 episodes? It is strange that there are even no quantitative results presented in the abstract. What do we better understand now about the formation, transport and concentrations of the extreme ozone than we did before what you did? What new insights can you provide? How would that help inform potential control measures or predictions? These are not well presented in the paper and in the abstract.

We agree with the reviewer. We acknowledge that the presentation of the results in the abstract and conclusions sections lacked the appropriate organization in the original version. Additionally, the contributing factors of the episodes were, in some cases, redundant as they expressed partly the same phenomena but in differentiated ways, adding complexity to the overall comprehension. In response to this feedback, we have thoroughly revised the presentation of these factors in both the abstract and concluding sections.

In the revised version, our focus has been on **redefining the common factors among the episodes**, and categorizing them into **four major factors** (instead of the previous seven), to enhance clarity and minimise redundancies.

In the revised version, the factors present in all the episodes are, briefly: (i) prior $O_3$ accumulation in coastal regions adjacent to Barcelona, (ii) the weekend effect, (iii) the presence of Tramontana

(Northern winds) meteorological conditions, and (iv) the multiregional convergence of polluted air masses. Within each factor, **we elaborate on how these factors may have influenced the episodes and incorporated specific differences** among the episodes (not shown here).

Furthermore, we have provided **concentration-based estimates of $O_3$ contributions from specific factors**. These estimates focus on the weekend effect and plausible $O_3$ contributions originating from different source areas, considering the convergence of contaminated air masses in the city. These quantitative results, now integrated into the revised conclusion section and abstract, enhance the contextual understanding of the findings and facilitate the accurate attribution of their significance. The detailed process for quantifying these contributions is detailed in the newly added Section S4 (supplementary material).

Moreover, based on the new redefinition of driving factors, **we have refined the presentation of information that can assist in forecasting the onset of these episodes, which can be valuable from a management standpoint**.

With these adjustments, we believe the presentation of our research results has improved and provides a better understanding of the processes involved in the occurrence of these kind of $O_3$ episodes in Barcelona.

2. The Introduction section is not well organized. Some contents are not directly related to this research topic. The main targets of the study are not well presented.

We have **reorganized parts of the Introduction** to focus on content directly relevant to the research, **reducing this section by approximately 15% from its original length**. The first paragraph now integrates information of formation regimes but shifts the focus to urban environments (as that of the study area) and the weekend effect, observed in all three episodes, making it pertinent to this investigation. Also, the wording has been improved, and the length shortened, improving the theoretical introduction to $O_3$. Furthermore, the section on epidemiological studies has been omitted as it lacked meaningful contributions. The paragraph introducing the phenomenology of episodes in the Western Mediterranean Basin (WMB) has been slightly refined, as it is necessary for understanding the complex phenomenology of $O_3$ in this region. The subsequent paragraph highlight scarcity of acute $O_3$ episodes in Barcelona, justifying the study's importance due to the high population exposed to very high $O_3$ concentrations during their occurrence. The following paragraph outlines the objectives.

Moreover, to improve the reader's comprehension of the structure of this paper, we added a brief section at the end. This segment is derived from the description of the subsections previously located at line 170 (Section 3.1) in the former manuscript (now relocated to the Introduction section), and supplemented with additional information.

Additionally, as suggested by the reviewer, we have **revised the primary objectives of the study, making them more concise**. One of the modifications involves removing a prior objective concerning the identification of common driving factors among episodes. This adjustment acknowledges that the identification of common factors among the episodes is indeed a key finding of our research that we did not know beforehand. Now the definition of the objectives reads:

*"This study aims to identify the underlying factors contributing to recent episodes of extreme O₃ concentrations in Barcelona by investigating the meteorological, transport, and formation mechanisms associated to these occurrences. Additionally, it aims to uncover novel insights to advance in the prediction of future events".*

3. Line 30, the major O3 precursors are NOx and VOCs; while the reactions of CH4 with radicals to generate O3 are very slow, suggest removing CH4 here. Additionally, the production of O3 is not only enhanced by high temperatures and low relative humidity, while solar radiation is one of the most important factors.

**We have removed CH₄ (and CO), leaving only the major precursors NOx and VOCs as suggested. We have also reformulated the sentences in this paragraph to highlight the importance of solar radiation in the formation of O₃.** Now, the section reads:

*"The formation of O₃ depends primarily on complex photochemical reactions between precursors (mainly nitrogen oxides (NOx) and non-methane volatile organic compounds (VOCs)), in the presence of sunlight. (…) The production of O₃ is enhanced by high solar radiation and temperatures, and low relative humidity (Monks et al., 2015), contributing to the occurrence of extreme O₃ events."*

4. Line 45, delete "characteristic".

To enhance readability, we have improved the sentence in addition to applying your suggestion:

*"In the western Mediterranean basin (WMB), many factors influence high O₃ concentrations (e.g., Millán et al., 1997, 2000; Gangoiti et al., 2001; Millán, 2014).; including  meteorological, climatic and topographic…"*

5. Line 71, "orography" should be a typo. ??

Thank you for the comment. In addition to incorporating your suggestion, we have identified that the sentences in this section were not properly formulated and did not convey our intended meaning effectively. We have now revised them to:

*"The intricate topography of the region, coupled with the prevailing summer conditions in the IP, influences local airflow patterns (Toll and Baldasano, 2000), and facilitates mesoscale circulations, including the northward channelling of sea/mountain breezes towards the Pyrenees (e.g., Barros et al., 2003; Diéguez et al., 2009)."*

6. Line 130-131, need to give references or websites of the RAMS and HYPACT models.

As suggested, we have added the references:

*"We conducted high-resolution backward and forward trajectory simulations using the mesoscale Regional Atmospheric Modelling System (RAMS) (http://www.atmet.com/software/rams_soft.shtml) and the HYbrid PArticle Concentration and Transport (HYPACT) model (http://www.atmet.com/software/hypact_soft.shtml)."*

7.  The anthropogenic emissions data used in this study are derived from EDGAR, which is published in Dec.2017. The emissions inventory is not UpToDate, which will surely introduce uncertainties.

We are fully aware that the anthropogenic emissions used in our study are derived from the EDGAR database, published in 2017, indicating that the emissions inventory is not up-to-date. It is true that this choice introduces certain uncertainties, as emissions are inherently variable over time. Thus **we have communicated this source of uncertainty in the manuscript** in the emissions description section. We have added the following to the paragraph describing emissions:

*"Emissions change significantly over time, but due to ease of calculation, and to conduct an analysis independent of emission variations, we have utilised the emissions inventory published in the year 2017, valid for monthly averages for 2010. The use of this inventory will introduce uncertainties but may not impact the main qualitative findings of our study."*

We think that the decision to consistently use the same emissions database provides significant advantages for our qualitative analysis. By not treating emissions as a variable, we can conduct a more focused analysis on regional transport, specifically, vertical and horizontal transport and recirculations.

8.  What kind of mechanisms are adopted in the CAMx model, and how do VOCs emissions speciated? This should be clearly given in the methodology section. Additionally, what about the model performance? The model needs to be evaluated before it can be used for further analysis.

We appreciate the referee's valuable input. Regarding the mechanisms adopted in the CAMx model and the speciation of VOCs emissions, while these details are not explicitly provided in the methodology section, we have incorporated relevant information through references to prior works (Torre-Pascual et al. 2023). To enhance clarity, **we have explicitly outlined the model mechanisms and VOCs speciation methodologies in the methodology section** of the manuscript.

*"We employed the gas-phase mechanism CB6r4 and used the SPECIATE tool (EPA, 2016) to speciate NOx and VOCs."*

On the other hand, we agree on the need to assess and report the performance of the model. Consequently, **we have added a new section in supplementary material (Section S6) titled "Performance of the photochemical model simulations" dedicated to the evaluation of the CAMx model**. In this section, we provide a summary of the effectiveness of the model in simulating $O_3$ concentrations during the three episodes, utilizing widely accepted validation metrics. Results show that the performance of the model aligns with findings from comparable studies, with Pearson Correlation Coefficients (r) slightly exceeding 0.7. Additionally, the negative values of Mean Bias indicate a consistent underestimation of O3 concentrations by the CAMx model. More details are presented in the supplementary section, providing a numerical and spatial distribution of selected statistical parameters for the three episodes.

References mentioned in this response:

EPA: U.S. Environmental Protection Agency. SPECIATE Version 4.5. Database Development Documentation. EPA/600/R-16/294, https://www.epa.gov/sites/default/files/2016-09/documents/speciate_4.5.pdf, 2016.

Torre-Pascual, E., Gangoiti, G., Rodríguez-García, A., Sáez de Cámara, E., Ferreira, J., Gama, C., Gómez, M. C., Zuazo, I., García, J. A., and de Blas, M.: Analysis of an intense O3 pollution episode in the Atlantic Coast of the Iberian Peninsula using photochemical modelling: characterization of transport pathways and accumulation processes, EGUsphere [preprint], https://doi.org/10.5194/egusphere-2023-387, 2023.

9. How do those factors (initial regional O3 accumulation; Tramontana winds; accumulation via vertical recirculation and horizontal circulation; convergence of polluted air masses from multiple sources; upper layer winds; weekend effect; abnormally high temperatures, etc) contribute to one specific O3 episode? Can the authors conclude with integrated systematic results? In fact, some of the factors are mixed and it is difficult to distinguish.

We recognise that presenting numerous mixed factors, which, in some cases are partially redundant, complicated their distinction and the understanding of their contribution to the episodes. Thus, we have partially addressed this feedback in our response to the previous comment #1. Additionally, in the modified concluding section of the revised version, as mentioned earlier, **we present in a systematic manner the four most important common factors leading to the occurrence of episodes**, providing clarity on how these factors contribute to the episodes. Furthermore, the inclusion of concentration-based estimates of specific $O_3$ contributions enhances the contextual understanding of the results and facilitates the appropriate attribution of certain factors.

10. It is hard to distinguish the commonalities and differences among all the influencing factors that cause the extreme O3 episodes. Most content are qualitative and descriptive, without further quantitative analysis.

Thank you for the feedback. We partially responded to this point in the previous comments (#1 and #9). **In the revised version the distinction between commonalities and differences is clear**: there are four main common driving factors, which are present in all the episodes. However, there are differences between episodes:

For example, the atmospheric mechanisms behind the prior accumulation of $O_3$ in the episodes (Factor 'i' in Response #1) differ among episodes. Another instance is that not all the episodes occurred during a "declared" heatwave, although temperatures were higher than usual in all cases. Another example of differences among episodes is that Tramontana conditions are present in all episodes (Factor 'iii' in Response #1), but the remote sources transported from southern France with the Tramontana winds, which finally impact in Barcelona, vary among episodes. One final illustration is that, during the convergence of multiregional sources occurring during the episode days (see Factor 'iv' in Response #1), area sources differ by episode, but there are always at least two of the three area sources concurrent. In essence, while the main contributing factors are common, they may exhibit internal differences.

Furthermore, we also included, as mentioned earlier, numerical estimates of some $O_3$ contributions to provide quantitative results, facilitating an attribution of their significance.

**Referee #2**
**https://doi.org/10.5194/egusphere-2023-2449-RC2**

The study entitled "Extreme ozone episodes in a major Mediterranean urban area" describes in detail 3 ozone pollution events in Barcelona that took place in 2015, 2018, and 2019, where the Europe information threshold (and alarm threshold on one occasion) where exceeded at several of the AQMSs in the metropolitain area.

The article goes into great detail through all three events, listing the onset conditions in terms of synoptic conditions, and compares meteorology and atmospheric composition observations with several model outputs and trajectory analysis. The authors conclude with a set of common conditions under which the three extreme ozone episodes happened.

I have found the study very informative and somehow suited to a diverse range of audiences with interests in meteorology, air quality, modeling, and observations. One limiting aspect is the repetitiveness in the description of the three episodes, though it does provide a systematic means of presenting all parameters that make up the analysis of each event.

We greatly appreciate the constructive feedback and have made efforts to address all the issues raised by the reviewer.

Indeed, one of the guidelines for this paper was to address a broad audience. We faced the challenge of thoroughly understanding three episodes using a variety of tools and successfully integrating this comprehensive analysis into a single paper. We maintained a consistent structure for each episode to facilitate a systematic understanding of the article. However, in some instances, as you have pointed out, this approach may create a sense of repetitiveness. This is particularly noteworthy (as findings suggest) because the factors generating the episodes are largely common among them, thus posing a challenge in terms of potential redundancy.

To address this issue, we made an effort to reduce the repetitiveness in the revised version. This involved a complete modification of the final section and abstract, as detailed below.

I would have liked to see a quantitative comparison of the model predictions - in relation to ozone levels in particular - with the observations. The article clearly states that the study provides a summary of conditions that permit such levels of ozone to build up in Barcelona as a way to better predict ozone pollution events, but the photochemical model systematically modelled elevated ozone for all three events - so I am currently not clear as to where the improved understanding needs to be inputted.

Following this comment, **we have included a model performance assessment by comparing model predictions and observations regarding O₃ levels at the surface level. To show the results, we added a new section in the supplementary material** (Section S6). The results suggest that the model performs well within an acceptable range.

Nonetheless, simulations represent just one of the multiple tools we employed. In this instance, our focus extends beyond the precise simulated concentrations; rather we seek to extract valuable insights to understand the processes contributing to the occurrence of the episodes, such as recirculations, subsidences, transport of polluted air masses, and other mechanisms related to the phenomenology of the episodes. Consequently, the improved understanding is attributed to the

interpretation of outcomes from all the tools we used, rather than solely relying on the model results.

As mentioned above, I struggle to understand what aspects of the conclusion need to be accounted by whom and what. The future work to follow on this study is also not clear. For example, the authors mention the weekend effects, which suggest the BMA is NOx limited - this has wide-ranging implications as anthropogenic emissions are expected to be further controlled/reduced.

We agree that the conclusions section of the original manuscript was not well-structured, among other issues, due to its repetitiveness. For this reason, a**s we also responded in the comment #1, we have completely revised this section. It now presents the results systematically, grouping the factors causing the episodes—essentially the same factors—into four major categories**. Within each of these groups, we have explained certain differences between the various episodes.

Furthermore, to enhance the value of the research results, **we have approximated certain $O_3$ contributions leading to abnormally high $O_3$ levels during the episodes**. The inclusion of these concentration-based estimates enhances the contextual understanding of the results and facilitates the appropriate attribution of their importance. For example, the weekend effect contribution has been estimated to be approximately 15 $\mu g \cdot m^{-3}$ in these episodes.

Additionally**, we have proposed future works in the conclusions section**, suggesting ideas for the potential expansion of our research.

A number of figures in the supplements are of poor to very poor resolution (S1.2a/b/c, S1.3, S2.2a), some captions are on a different page, and S2.2 is missing its (C). Please check all figures in the main document and supplements and improve the resolution or clarity of dates, axis, and colorbars where necessary.

Thank you for your comment. **We have carefully reviewed all the figures** in both the main manuscript and supplements and improved their resolution and readability. We ensured that captions are on the same page of the corresponding figures, and added the missing "(C)" in the figure S2.2.

A few specific issues are listed below:

L51: "Other basins in Spain". Spain had yet been mentioned thus far in the intro.

Thank you for this comment. We improved the sentence:

*"...regional transport from other air basins  and other parts of Europe..."*

L86: typo with reference

Many thanks. We added the missing parenthesis:

*"...northern and northwestern areas (Querol et al., 2016)."*

L132: "Centred on Barcelona" maybe?

Thank you for the comment; **we used the preposition suggested**:

*"...We used three nested square domains, centred on Barcelona ..."*

L179: This is the only mention of ppb in the study, while everything else is in ug/m3.

We appreciate the reviewer's comment. There are additional mentions of ppb in the manuscript, specifically in the 3 captions of the figures showing ERA-INTERIM reanalyses (Figures 2, 6 and 10), the cross section modelling outputs (figures S1.6 and S2.5), and in the text line 407. We have employed ppb units, as these prove more pertinent for concentrations at higher altitudes (not only at the surface). **Thus, to improve the text considering your suggestion, we specified the concentrations in "µg·m$^{-3}$" after the original "ppb" mentions**:

L211: *"The ERA-INTERIM reanalysis indicated a 50 ppb (~100 µg·m$^{-3}$) O$_3$ accumulation..."*

L439: *"...possibly contributing to the accumulation of O$_3$ (55–60 ppb, ~110–120 µg·m$^{-3}$), as also shown in the ERA-INTERIM reanalysis..."*

We also modified the captions for Figures 2,6 and 10, where we added the conversion from ppb to µg·m$^{-3}$. Here we show the example of Figure 2:

*"... (bottom) ERA-INTERIM (ECMWF) reanalysis (0.75° resolution) of O$_3$ concentration (ppb, contour-lines; **1 ppb ≈ 2 µg·m$^{-3}$ at sea level**), ..."*

And also for Figures S1.6 and S2.5:

*"... with vertical winds. **Concentrations are shown in ppb since they are altitude independent (1 ppb ≈ 2 µg·m$^{-3}$ at sea level).**"*

L228: I think you meant S2.2b. Also, I cannot see a reduction in solar radiation in the figures (maybe a minimal reduction in S2.2d)…"

We appreciate the reviewer's comment. However, we believe the reviewer is referring to line 328. Certainly, the reference to Figure 2.2a is not correct. The reference should be to Figure S2.2 in its entirety (no specific subsection), as in the text, we make mention of meteorological parameters distributed across various subsections, thus we removed the "a" in "2.2a".

With respect to the variation in solar radiation in the days following the episode, indeed, it is limited. We have calculated a 11% decrease in solar radiation compared to the average of days 6 and 7, in relation to the episodic days 4 and 5. It is indeed a subtle reduction that may be challenging to discern in the graph; hence, we have included this quantification in the text for clarification. With these changes, the text now reads (L361):"

*"...an increase in relative humidity (+30–35%), and weaker solar radiation **(–11%)**, compared to the episode days **(Fig. S2.2).**"*